# Determinants of Carotid Wall Echolucency in a Cohort of European High Cardiovascular Risk Subjects: A Cross-Sectional Analysis of IMPROVE Baseline Data

**DOI:** 10.3390/biomedicines12040737

**Published:** 2024-03-26

**Authors:** Beatrice Frigerio, Daniela Coggi, Alice Bonomi, Mauro Amato, Nicolò Capra, Gualtiero I. Colombo, Daniela Sansaro, Alessio Ravani, Kai Savonen, Philippe Giral, Antonio Gallo, Matteo Pirro, Bruna Gigante, Per Eriksson, Rona J. Strawbridge, Douwe J. Mulder, Elena Tremoli, Fabrizio Veglia, Damiano Baldassarre

**Affiliations:** 1Centro Cardiologico Monzino, IRCCS, 20138 Milan, Italy; beatrice.frigerio@cardiologicomonzino.it (B.F.); daniela.coggi@cardiologicomonzino.it (D.C.); alice.bonomi@cardiologicomonzino.it (A.B.); mauro.amato@cardiologicomonzino.it (M.A.); nicolo.capra@cardiologicomonzino.it (N.C.); gualtiero.colombo@cardiologicomonzino.it (G.I.C.); daniela.sansaro@cardiologicomonzino.it (D.S.); alessio.ravani@cardiologicomonzino.it (A.R.); 2Foundation for Research in Health Exercise and Nutrition, Kuopio Research Institute of Exercise Medicine, 70100 Kuopio, Finland; kai.savonen@uef.fi; 3Department of Clinical Physiology and Nuclear Medicine, Kuopio University Hospital, 70210 Kuopio, Finland; 4INSERM, Unité de Recherche sur les Maladies Cardiovasculaires, le Métabolisme et la Nutrition, ICAN, Sorbonne Université, F-75013 Paris, France; philippe.giral@aphp.fr (P.G.); antonio.gallo@aphp.fr (A.G.); 5Lipidology and Cardiovascular Prevention Unit, Department of Nutrition, APHP, Sorbonne Université, Hôpital Pitié-Salpêtrière, F-75013 Paris, France; 6Internal Medicine, Angiology and Arteriosclerosis Diseases, Department of Medicine and Surgery, University of Perugia, 06129 Perugia, Italy; matteo.pirro@unipg.it; 7Department of Medicine Solna, Division of Cardiovascular Medicine, Karolinska Institutet, Stockholm, Karolinska University Hospital, 17177 Solna, Sweden; bruna.gigante@ki.se (B.G.); per.eriksson@ki.se (P.E.); rona.strawbridge@glasgow.ac.uk (R.J.S.); 8School of Health and Wellbeing, University of Glasgow, Glasgow G12 8TB, UK; 9Health Data Research UK, Glasgow G12 8TA, UK; 10Department of Internal Medicine, University Medical Center Groningen, University of Groningen, 9700 RB Groningen, The Netherlands; d.j.mulder@umcg.nl; 11Maria Cecilia Hospital, GVM Care & Research, 48033 Cotignola, Italy; etremoli@gvmnet.it (E.T.); fveglia@gvmnet.it (F.V.); 12Department of Medical Biotechnology and Translational Medicine, Università degli Studi di Milano, 20129 Milan, Italy

**Keywords:** echolucency, gray-scale median, atherosclerosis, carotid plaque, intima–media thickness, cardiovascular, risk factors

## Abstract

Echolucency, a measure of plaque instability associated with increased cardiovascular risk, can be assessed in both the carotid plaque and the plaque-free common carotid intima–media (IM) complex as a gray-scale median (plaque-GSM and IM-GSM, respectively). The impact of specific vascular risk factors on these two phenotypes remains uncertain, including the nature and extent of their influence. This study aims to seek the determinants of plaque-GSM and IM-GSM. Plaque-GSM and IM-GSM were measured in subjects from the IMPROVE study cohort (aged 54–79, 46% men) recruited in five European countries. Plaque-GSM was measured in subjects who had at least one IMT_max_ ≥ 1.5 mm (*n* = 2138), whereas IM-GSM was measured in all subjects included in the study (*n* = 3188). Multiple regression with internal cross-validation was used to find independent predictors of plaque-GSM and IM-GSM. Plaque-GSM determinants were plaque-size (IMT_max_), and diastolic blood pressure. IM-GSM determinants were the thickness of plaque-free common carotid intima–media complex (PF CC-IMTmean), height, systolic blood pressure, waist/hip ratio, treatment with fibrates, mean corpuscular volume, treatment with alpha-2 inhibitors (sartans), educational level, and creatinine. Latitude, and pack-years_code_ were determinants of both plaque-GSM and IM-GSM. The overall models explain 12.0% of plaque-GSM variability and 19.7% of IM-GSM variability. A significant correlation (r = 0.51) was found between plaque-GSM and IM-GSM. Our results indicate that IM-GSM is a weighty risk marker alternative to plaque-GSM, offering the advantage of being readily measurable in all subjects, including those in the early phases of atherosclerosis where plaque occurrence is relatively infrequent.

## 1. Introduction

The potential of an atherosclerotic lesion to produce a cardiovascular (CV) event is determined not only by its size, but also by its susceptibility to rupture. The latter, in turn, is often associated with its tissue composition [1]. Tissue composition can be estimated in terms of echolucency using high-resolution B-mode ultrasound; an ultrasonographic technique normally used to measure plaque-size and intima–media thickness (IMT) of superficial arteries (e.g., carotids). Specifically, the echolucency of the plaque can discriminate between stable and unstable lesions [2,3,4,5]. Echolucent plaques, which are more prone to rupture [6], are characterized by high lipid content, high macrophage content [6,7], and, sometimes, intraplaque hemorrhage [6]. By contrast, echorich plaques are substantially more stable and are associated with a high content of fibrous tissue [8]. Plaque echolucency correlates with many vascular risk factors (VRFs) including: lipids [9,10], insulin resistance [11], smoking [12], body mass index [13], markers of oxidative stress [9], and others [14]; and it is a significant predictor of coronary- [15] and cerebro-vascular events [15,16].

Echolucency can be measured in terms of gray-scale median (GSM), not only at the plaque level (plaque-GSM), but also at the level of the intima–media (IM) complex of the common carotid arteries in plaque-free areas (IM-GSM). Similar to plaque-GSM, IM-GSM has also demonstrated its role as a CV risk marker [17,18]; in fact, it has been linked to VRFs in both the general population [11,12,19] and in low-risk populations [13,20]. Furthermore, it has been employed to evaluate the impact of antiatherosclerotic medications [8].

Whether plaque-GSM and IM-GSM have comparable clinical significance is still unknown. Indeed, the majority of the studies published to date have been carried out considering either plaque-GSM or IM-GSM [5,13]. Up to now, only a few studies [17,18,21,22,23,24,25,26,27] have measured both variables in the same subjects, but none of them have investigated whether one can be considered a proxy (i.e., an index) of the other, or whether the two phenotypes are associated with the same VRFs or not. Since the histopathological characteristics of the two variables appear to be different [28], it is plausible to assume that their clinical significance may also be different. To gain further understanding on this matter, we have conducted a comprehensive analysis of the primary factors influencing both plaque-GSM and IM-GSM in European individuals who have been classified as having a high CV risk due to the presence of at least three VRFs.

## 2. Materials and Methods

### 2.1. Participants

This study was conducted utilizing the dataset and imaging repository of IMPROVE [29]: a multicenter, observational, and longitudinal study that was specifically designed to identify novel indicators of vascular events [29].

Detailed information regarding the eligibility criteria for participant enrollment, study objectives, methodologies, and baseline characteristics of the subjects have been previously described [29]. Briefly, the original IMPROVE cohort comprised 3703 participants (48% male, aged 54–79 years) who were asymptomatic for CV and cerebrovascular diseases at the time of enrollment, and had adequately visualized carotid artery walls. Participants were enrolled in seven centers in five European countries: Finland, two centers (University of Kuopio; *n* = 515 and Kuopio Research Institute of Exercise Medicine; *n* = 533); Sweden (Karolinska Institutet, Stockholm; *n* = 532); the Netherlands (University Medical Center of Groningen and Isala Clinics of Zwolle; *n* = 527); France (Groupe Hôspitalier Pitié-Salpetriere, Paris; *n* = 501); and Italy, two centers (University of Milan (*n* = 553) and University of Perugia (*n* = 542)). The Institute of Public Health and Clinical Nutrition at the University of Eastern Finland in Kuopio (Kuopio-1) has withdrawn permission to use its data starting from 2023. Consequently, such data have been deleted from the dataset, resulting in 3188 participants being included in the analyses.

The IMPROVE study adheres to the regulations of good clinical practice and with the ethical principles of the Helsinki Declaration, and was approved by local ethics committees. All participants provided written informed consent.

### 2.2. Quantitative Assessment of Carotid Artery Echolucency

The protocol for carotid ultrasound has previously been described [30]. Briefly, the far walls of common carotids (CC), bifurcations (Bif), and internal carotid arteries (ICA) were visualized from three angles (anterior, lateral, and posterior) and logged on super Video Home System (sVHS) videotapes. All carotid IMT ultrasonographic variables measured in the IMPROVE study were formerly defined [29]. Variables of interest for the present study are plaque-free (PF) CC-IMT_mean_, IMT_max_, and plaques. PF CC-IMT_mean_ is the average of right and left intima–media (IM) thickness measured in PF areas of the 2nd cm of CC arteries. IMT_max_ is the highest of the IMT_max_ values detected in ICA, Bif, and CC of right and left carotids. A plaque is an atherosclerotic lesion with an IMT_max_ ≥ 1.5 mm.

Ultrasonographic scans were subsequently digitized using a Medical Digital Recording device (SONY HD Video Recorder—HVO-500MD (Surgical Version), SONY Europe). Selected frames were then loaded into a dedicated semi-automatic software (MIA Carotid Analyzer, Coralville, IA, USA, version 6.7.5) for echolucency analysis. The software carries out a video-densitometric analysis of a ROI (region of interest) and automatically provides GSM, i.e., 50th percentile of the pixel distribution in a gray-scale which ranges from zero (the darkest tone) to 255 (the brightest tone). Echogenicity of both carotid plaques (plaque-GSM) and PF CC intima–media complex (IM-GSM) were quantified. To assess plaque-GSM or IM-GSM, an ROI was manually positioned around the segment to be analyzed and the software automatically calculated the GSM level of the pixels within the ROI. The main limitation of these techniques is that GSM assessment is influenced by the ultrasonographic device used and by image acquisition settings (gain and brightness). To overcome these limits, seven identical machines (Technos, Esaote, Genoa, Italy) equipped with the same probe (5 to 10 MHz linear array probe) were adopted. In addition, all the devices have been prepared with exactly the same settings and precision in size measurements was checked versus a phantom at baseline and after 1 year. In addition, the measurement software underwent recalibration for each individual image. For this purpose, in each subject the blood-echolucency measured in the darkest section of the vessel lumen (ranging from 0 to 37) was designated as the “reference black”, while the echolucency measured at the lightest part of the adventitia (ranging from 104 to 255) was regarded as the “reference white”. In cases where the automatic tracing of plaque-edges or IM-edges did not meet the operator’s visual examination, the traced profile could be manually corrected.

Low GSM values represent echolucent lesions (examples in Figure 1a and Figure 2a), whereas high GSM values represent echogenic lesions (examples in Figure 1b and Figure 2b).

Plaque-GSM was measured only in lesions with an IMT_max_ ≥ 1.5 mm. When plaques were present in both carotids, plaque-GSM was obtained by considering the darkest one among the two thickest plaques detected on far-walls of right and left carotid arteries, regardless of their location on CC, Bif, or ICA. IM-GSM was computed by averaging the GSM obtained from the lateral projection of the 2nd cm of the far-wall of left and of right CC arteries in PF areas (i.e., PF CC-IMT_mean_).

The reproducibility of measurements for plaque-GSM and IM-GSM was assessed in 138 subjects who underwent initial scanning during the baseline visit and a subsequent scan two weeks later. The mean absolute differences (±SD) of GSM values in the comparison of repeated scans were 5.3 ± 6.2 for plaque-GSM and 4.8 ± 5.7 for IM-GSM, respectively. The intraclass correlation coefficients for the corresponding variables were 0.75 and 0.83, respectively.

### 2.3. Statistical Analysis

Each variable with normal distribution was described as “mean ± SD”. Variables with skewed distribution were described as “median (inter-quartile range)” and log-transformed before analysis. Categorical variables were described as “frequency (percentage)”. Pack-years_code_ was obtained by dividing the pack-years continuous variable into 4 categories: 0 (never smokers), 1, 2, and 3 (first, second, and third tertile of pack-years, respectively). Group differences according to quartiles of GSM were assessed by ANOVA (with Bonferroni post hoc test for multiple comparisons) for numerical variables with normal distribution, by Kruskal–Wallis for variables with skewed distribution, and by χ^2^-test or Fisher’s exact test for categorical variables.

Despite initial calibration, black and white values were still potential confounders in most of the analyses considered. Hence, plaque-GSM and IM-GSM were a priori adjusted for black and white values before the analysis. Independent predictors of plaque-GSM and IM-GSM were identified through multiple linear regression analysis considering all the variables listed in Appendix A. To limit spurious associations, the reliability and consistency of the identified subset of predictors were tested with a cross-validation iteration procedure. At each stage, the dataset was randomly divided in two halves. Predictors were selected in the first half (training set) and the resulting model was tested for significance in the second half (testing set). The procedure was repeated 200 times with different random splits. Since a consensus on a specific threshold was not available, we decided to consider a predictor as “reliable” only when it was chosen and validated by the cross-validation procedure in at least 70% of cases. The percent of GSM variation explained by the identified predictors was quantified by partial correlation analysis. The variables identified in the previous step were included in a logistic regression aimed at assessing the area under the ROC curves (AUC), to determine their potential in predicting plaque-GSM or IM-GSM. As no established cutoff is reported in the literature for plaque-GSM or IM-GSM, they were both dichotomized as below or above their respective medians. The relationship between plaque-GSM and IM-GSM was computed using the Pearson’s correlation analysis. Logistic regression was utilized to investigate the ability of both plaque-GSM and IM-GSM in predicting a high atherosclerotic burden; defined as a carotid IMT_max_ in the upper quartile. In Appendix A
*p*-values < 0.0007 (threshold according to Bonferroni correction for 73 independent comparisons) were considered as significant. All tests were two-sided. Statistical analysis was carried out with SAS statistical package (SAS Institute Inc., Cary, NC, USA) version 9.4.

## 3. Results

### 3.1. Associations between Plaque-GSM, IM-GSM and Participants’ Characteristics

Plaque-GSM was measured in 2138 participants having at least one lesion with an IMT_max_ ≥ 1.5 mm. IM-GSM was measured in all the 3188 participants. The GSM distribution of plaque and IM are shown in Appendix A. On average, plaque-GSM (30.6 ± 9.9; *n* = 2138) was significantly lower (darker) than IM-GSM (43.8 ± 11.7; *n* = 3188); *p* < 0.0001.

The characteristics of IMPROVE participants stratified by plaque-GSM or IM-GSM quartiles are shown in Appendix A. Univariate associations with respective standardized beta coefficients and *p* trends are also shown in the same tables. In order to select the variables to be included in the multiple linear regression carried out to identify the plaque-GSM or IM-GSM independent determinants, the robustness of the univariate association observed in Appendix A was tested with a cross-validation procedure (Appendix A). In such analysis, latitude and pack-years_code_ were identified as robust predictors of both plaque-GSM and IM-GSM because they were selected and confirmed by the cross-validation procedure at least 70% of the time (Appendix A). According to the same criteria, IMT_max_ and PF CC-IMT_mean_ were identified as robust determinants for plaque-GSM (Appendix A) and for IM-GSM (Appendix A), whereas diastolic blood pressure (DBP) was identified as robust determinant of plaque-GSM only (Appendix A); and waist/hip ratio, use of fibrates, creatinine, educational level (study years), height, mean corpuscular volume (MCV), use of sartans, and systolic blood pressure (SBP) were identified as determinants of IM-GSM only (Appendix A). In summary, the variables considered sufficiently robust to be included in the multiple regression analysis required to identify independent predictors were four (latitude, IMT_max_ (quartiles), pack-years_code_ and DBP) for plaque-GSM, and eleven (latitude, PF CC-IMT_mean_ (quartiles), pack-year_code_, waist/hip ratio, fibrates, creatinine, education level, height, MCV, sartans and SBP) for carotid IM-GSM.

### 3.2. Independent Determinants of Carotid Plaque-GSM and IM-GSM

In multiple regression analysis (Table 1 and Table 2), latitude, and wall thickness (IMT_max_ or PF CC-IMT_mean_) were negatively associated with both plaque-GSM and IM-GSM.

Pack-years_code_, and blood pressure were positively associated with both plaque-GSM and IM-GSM. Waist/hip ratio, the use of fibrates or sartans, MCV, and educational level were negatively associated with IM-GSM only. Height and creatinine were positively associated with IM-GSM.

The 12.0% of plaque-GSM variability and the 19.7% of IM-GSM variability were explained by the variables included in the models (Figure 3a and Figure 3b, respectively).

ROC curves analysis (Appendix A) confirmed that the set of selected variables better predicted IM-GSM than plaque-GSM (AUC = 0.742, 95% CI 0.725; 0.760, vs. 0.681, 95% CI 0.659; 0.704, respectively, *p* < 0.0001 for the comparison of the two AUCs).

### 3.3. Relationship between Plaque-GSM, IM-GSM and Atherosclerotic Burden

Plaque-GSM and IM-GSM correlation was highly significant (r = 0.51; *p* < 0.0001). To further investigate the relative roles of plaque-GSM and IM-GSM in predicting a high atherosclerotic burden (evaluated in terms of IMT_max_ in the upper quartile), we stratified the study population in quartiles of plaque-GSM and quartiles of IM-GSM. Appendix A shows the Odds Ratio (OR) with the 95% CI of IMT_max_ computed after data adjustment for latitude, sex, age, educational level, pulse pressure, and pack-years_code_ (i.e., for the strongest determinants of IMT_max_) [30]. Figure 4 shows that the lower the plaque-GSM and the higher the IM-GSM, the greater the OR of having an IMT_max_ in the upper quartile of its distribution (≥2.5 mm).

As a consequence, the concomitant presence of low plaque-GSM (echolucent plaques) and high IM-GSM (echogenic IM) yielded a marked increase in the OR. The OR for high IMT_max_ associated with one GSM quartile increase was 0.79 (95% CI 0.71; 0.86) for plaque and 1.12 (95% CI 1.02; 1.24) for IM. The relationship between participants’ atherosclerotic burden and plaque-GSM appeared to be linear, whereas the relationship between participants’ atherosclerotic burden and IM-GSM was compatible with a U-shape (Figure 4). This is confirmed also when data were stratified for just IM complex (Appendix A).

The association depicted in Figure 4 between the OR for having a high atherosclerotic burden and GSM, which is opposite when considering plaque or intima–media, is further reinforced by the same analysis conducted after stratifying by sex, age, and baseline CV risk (Appendix A). Indeed, even after these stratifications, plaque-GSM continues to exhibit negative associations, while IM-GSM maintains positive associations.

## 4. Discussion

To the best of our knowledge, this study represents the first comparison, within the same group of subjects, of factors influencing the echolucency of both carotid plaque and the common carotid intima–media complex. The main result was that there is some overlap between determinants associated with the two measures, suggesting the lesions evaluated (IMT_max_ for plaque-GSM and PF CC-IMT_mean_ for IM-GSM) may affect not only the composition of plaque, but also the composition of the IM complex. Despite these similarities, our study also shows some interesting dissimilarities between plaque-GSM and IM-GSM. Firstly, the overall regression model explains 12% of plaque-GSM and ~20% of IM-GSM variability, suggesting that traditional VRFs predict IM complex composition better than plaque composition. This is not surprising, since many more biochemical and metabolic pathways are involved in plaque composition than in IM complex composition [28]. Indeed, in a study comparing B-mode ultrasound measurement and histopathological features it has been shown that the predominant component of IM complex was fibrosis, with a minor presence of fatty necrotic debris and/or calcification [28]. Conversely, rare fibrosis and greater complexity was often observed in plaque composition, with frequent atheromatous debris, high content of cholesterol, recent hemorrhage, and superficial thrombosis [28].

Another noteworthy difference was that participants’ atherosclerotic burden was negatively associated with plaque-GSM but positively associated with IM-GSM (Figure 4). The relationship between the atherosclerotic burden and IM-GSM was U-shaped. The U-shape, only minimally appreciable in Figure 4, is much more evident when data were stratified just for IM complex (Appendix A). Such a result is in line with that of Wohlin et al. [17], who found a similar U-shaped relationship between risk of CV event and IM-GSM. Such a finding suggests that both a too-echolucent and a too-echogenic IM-complex might be harmful. The biological mechanisms that might explain such a U-shape have not been defined yet. In his article, Wohlin hypothesized that subjects with a high IM-GSM may exhibit a more pronounced presence of collagen [17]. If this is the case, the echolucent IM-complex would suggest individuals with a more unstable form of atherosclerosis, whereas those with an echogenic IM-complex would suggest individuals with stiffer arteries. According to these results, measuring IM-GSM in addition to plaque-GSM could provide complementary information for CV risk stratification. Such complementarity may be the consequence of the fact that echolucency and echorichness have different histological contexts, and consequently they may also have different pathophysiological significance.

### 4.1. Positive and Negative Predictors of GSM

Latitude was the strongest independent determinant of low plaque-GSM (echolucent plaque) in our population of European subjects, accounting for 8.7% of the total variance. Most investigations addressing the relationship between echolucency and VRFs were single-nation studies [10,18,31,32] and the effect of latitude was unassessable. Our study, including nations with a north-to-south distribution, has shown the effect of latitude on carotid plaque-GSM. This is consistent with results previously observed in the same population, in which latitude was the strongest determinant of another marker of atherosclerosis, i.e., the carotid IMT_max_ [30]. In European countries, an analogous geographical gradient was also reported for the incidence of CV diseases [33]. Latitude was also a strong determinant of IM-GSM, accounting for 11.1% of the total variability. This suggests that the effect of latitude, reflecting differences across countries in terms of lifestyle, diet, socioeconomic status, burden of VRFs, etc. [30], is more pronounced in early rather than in advanced stages of atherosclerosis.

Other significant determinants of both carotid plaque-GSM and IM-GSM were pack-years_code_ and blood pressure. The impact of smoking as a substantial subclinical atherosclerosis risk factor has been well established in several studies demonstrating an association with elevated carotid IMT [34] and with its progression [35]. However, the effect of smoking on plaque composition has not been fully elucidated. In our study, lifelong tobacco exposure, as quantified by pack-years, was associated with high plaque-GSM and high IM-GSM. This positive association is in line with the results of two large studies; i.e., the MESA study (6384 women and men, aged 45–84, free from clinical CV disease) [36] and the PIVUS study (1016 subjects, aged 70 years, population-based health survey) [37]. In the first, pack-years were significantly associated with echorich (high-GSM) plaques [36]. In the second, significantly higher pack-years were related to echorich plaques [37]. Different results were obtained by other authors showing no significant associations [20,38] or even inverse association with carotid plaque-GSM [39,40] or with carotid IM-GSM [20,41,42]. Still different results were obtained in the Northern Manhattan Study (1743 stroke-free participants of urban multi-ethnic population, age 65.5 ± 8.9 years), in which a non-linear relationship between cigarette smoking and plaque echogenicity was detected, with individuals who actively smoked having more soft or calcified plaques when compared with never smokers [32]. Furthermore, in a cohort study conducted within a Japanese community, cigarette smoking emerged as an independent risk factor of high-risk atheroma, i.e., of lesions with mainly hypoechoic features and/or ulceration [43]. The findings of this investigation suggest that the association between cigarette smoking and the susceptibility to CV events could be attributed to the heightened risk of plaque vulnerability, potentially attributable to augmented arterial wall stiffness and the inflammatory response elicited by cigarette smoking [43]. However, the rise in carotid artery echogenicity reported in other studies, which may indicate the onset of a fibrotic or sclerotic process triggered by components found in tobacco smoke [41], does not inherently imply a decreased CV risk. Indeed, within a sample of elderly men from a community-based cohort, both high and low levels of echogenicity were associated with heightened CV mortality [17]. Thus, further research is warranted to elucidate the intricate mechanisms underlying the influence of cigarette smoking on the composition of arterial plaques.

Few studies have investigated the association between blood pressure and carotid echolucency [37,44,45]. SBP was inversely related to IM-GSM in healthy postmenopausal women [44] and to plaque-GSM in elderly subjects [37,45] and in carriers of stenotic plaques [10]. Furthermore, long-term treatment with beta-blockers (more than 6 months) was associated with higher plaque-GSM; suggesting a positive effect of hypertension control [46]. In our study, by contrast, we found no association between SBP and the plaque-GSM and even an association in the opposite direction when IM-GSM was considered. However, it should be emphasized that SBP in our study was only moderately high (142 ± 19 mmHg) and was lower than that observed in the PIVUS (152 ± 23 mmHg) and Tromsø studies (157 ± 24 mmHg). Furthermore, our results could be influenced by the high percentage of participants treated with antihypertensives (beta-blockers ~22%, diuretics ~25%, ACE inhibitors ~19%, calcium antagonists ~16%, alpha-2 inhibitors ~14%). Indeed, even though the beta coefficient for SPB versus IM-GSM was minimally modified by further adjustment for antihypertensive treatment (Appendix A), a lower coefficient was still observed in treated versus untreated subjects. Similar results have been obtained when the analyses shown in Appendix A were repeated considering lipid lowering or blood glucose lowering drugs (Appendix A).

Waist/hip ratio was only a determinant of low IM-GSM. These findings are consistent with those of Mitchell et al. [20], Genkel et al. [47], Sarmento et al. [48], and Lind et al. [11], who reported an inverse relationship between IM-GSM and waist circumference [20,47,48] or waist/hip ratio [11], respectively. Other studies have demonstrated a significant association between markers of obesity and GSM [11,13,41,42,44]. For instance, the METEOR study [13], The Women’s Interagency HIV Study [41], a population-based sample of elderly men [11], a smoking cessation trial [42], and Healthy Postmenopausal Women [44] have all reported an inverse association between increased BMI and IM-GSM. The link between GSM and obesity may be related to the dysregulated secretion of adipokines by adipose tissue, which is recognized as one of the key pathological features of obesity [49]. Specifically, previous studies have shown that adiponectin, a beneficial adipokine that prevents lipid accumulation in the vascular wall [27], is downregulated in individuals with obesity [49]. Low levels of adiponectin were found to be correlated with the presence of lipid-rich plaques (low-GSM) in the coronary arteries of patients undergoing intravascular ultrasound [50]. Reduced levels of adipokine were also correlated with the echolucent plaques presence (low-GSM) in the carotids of diabetic women [51]. Finally, this correlation was observed not only for carotid plaques but also for carotid IMT. In the PIVUS study, the lower the adiponectin levels, the lower the GSM of both IMT and plaques [27]. In our study, we observed, as expected, a positive and highly significant association between adiponectin and both plaque-GSM and IM-GSM in the univariate analysis; however, such correlations were not confirmed in the cross-validation analyses and, as such, adiponectin was not included in the multiple regression analyses. Forcing adiponectin in the model did not substantially change the relationship between plaque-GSM or IM-GSM and waist/hip ratio (Appendix A), thus suggesting that such a relationship was independent of this variable. In contrast, the fact that the highly significant correlation between both plaque-GSM or IM-GSM and adiponectin observed in the univariate analyses was no longer significant in the multivariate analyses suggests that, at least in our study, such correlation is spurious and dependent on another not yet considered variable.

Low educational level, a consistent index of low socioeconomic status (SES), was previously associated with carotid atherosclerosis [52,53,54], VRFs and mortality [55,56]. Regarding arterial wall composition, the relationship between echolucency and SES, or between echolucency and indexes of SES such as educational level or lifelong occupation has never been investigated, and our study is the first to show that such an association exists also with arterial wall composition, at least considering the GSM of IM complex. The fact that educational level negatively associates with GSM is intriguing and may indicate that low SES exerts its detrimental effect mainly on IM thickness, whereas the IM composition even seems to be beneficially affected. If this apparent and paradoxical protective effect is real or derived from a spurious correlation with a not yet considered variable remains to be clarified in other studies.

Finally, it may seem surprising that the association with two of the most important predictors of CV disease and of subclinical atherosclerosis, namely age and sex, was not confirmed for either plaque-GSM or IM-GSM. However, this lack of association can be explained by the strong independent relationship existing between lesions thickness (IMT or plaque) and age and sex on the one hand, and with echolucency (GSM) on the other. When lesion size (IMT or plaque) is included in a multivariable model, the relationship between age and sex and GSM, due to the aforementioned relationships, tends to disappear.

### 4.2. Study Limitations

The present study has some limitations. (1) The IMPROVE participants were recruited based on the presence of a minimum of three VRFs, thus these findings must be extrapolated cautiously to general European populations or to subjects with fewer than three VRFs. However, in a subgroup analysis, the relationship between both plaque-GSM and IM-GSM was very similar in low- and high-risk subjects (Appendix A). (2) The whole model accounts for 12% of plaque-GSM variability and for ~20% of IM-GSM variability, suggesting suboptimal performance and potential unexplored determinants. (3) Since GSM was measured on pre-recorded images, for plaques with very low GSM the reader could not take advantage of color or power doppler imaging to identify the exact borders of carotid plaques. (4) Plaque composition and IM composition was evaluated only in terms of GSM, but other variables such as entropy [57], pixel distribution analysis [58], integrated backscatter ultrasound analysis [59], or shear wave elastography [60] can help to better identify unstable carotid plaques.

## 5. Conclusions

This study provides the first demonstration, in a large European cohort classified as at high risk for CV disease due to the presence of at least three VRFs, that several determinants of plaque-GSM are also determinants of IM-GSM. This result suggests that IM-GSM, which reflects the carotid arterial wall composition, is an important and easily measurable marker that can be obtained in all subjects. This could be particularly relevant not only for subjects at an early stage of atherosclerosis, but also for subjects at high CV risk that, even in the presence of three or more VRFs, do not develop overt plaques.

Our study also demonstrated an independent relationship between latitude and carotid echolucency, confirming the north-to-south gradient previously documented for carotid IMT. The fact that it is independent of other VRFs reinforces our previous suggestion that among mechanisms at the basis of this geographical gradient are heritable factors predisposing to (in the north) or protecting from (in the south) more complicated atherosclerotic lesions.

## Figures and Tables

**Figure 1 biomedicines-12-00737-f001:**
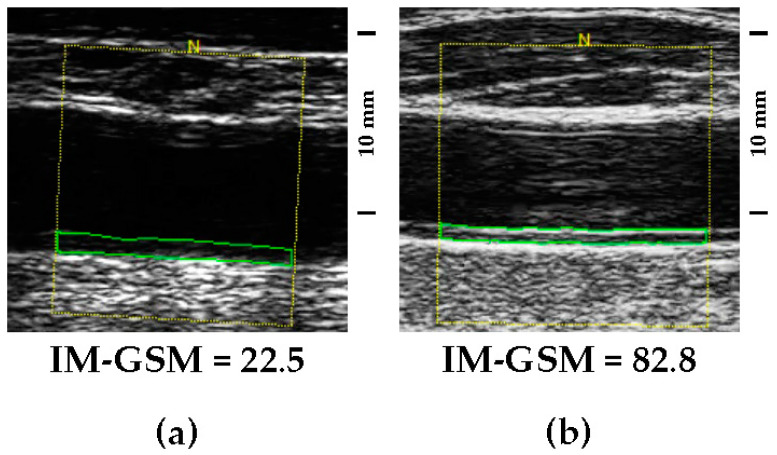
(**a**) Didactic example of an echolucent intima–media complex (low IM-GSM). (**b**) Didactic example of an echogenic intima–media complex (high IM-GSM). Both images were detected in the 2nd cm of two common carotid arteries visualized by using the same ultrasonographic gain setting. Images reported are the most representative, and were chosen from a shortlist of about 10 random images. The yellow frame identifies the region of interest, while the green frame identifies the intima-media complex.

**Figure 2 biomedicines-12-00737-f002:**
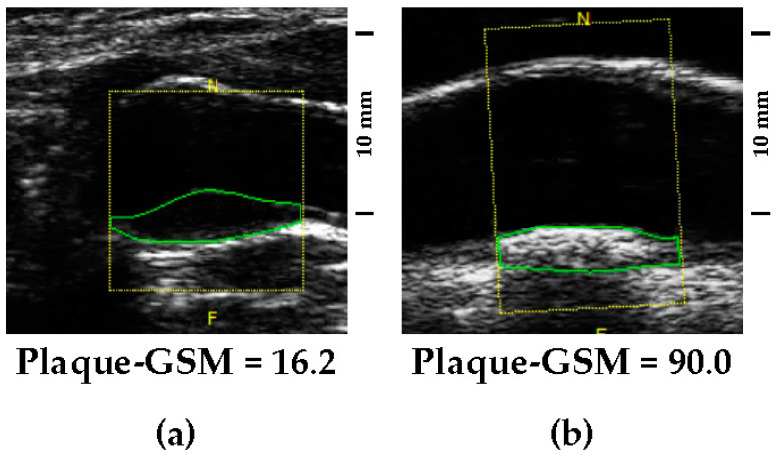
(**a**) Didactic example of an echolucent plaque (low plaque-GSM). (**b**) Didactic example of an echogenic plaque (high plaque-GSM). Both images were detected in carotid arteries visualized by using the same gain setting. Images reported are the most representative, chosen from a shortlist of about 10 random images. The yellow frame identifies the region of interest, while the green frame identifies the atherosclerotic lesion.

**Figure 3 biomedicines-12-00737-f003:**
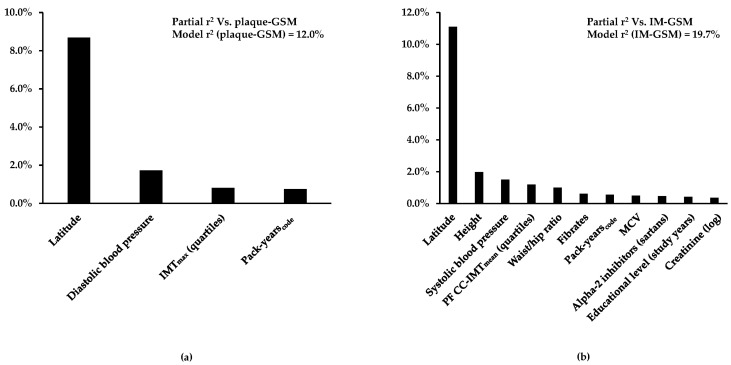
(**a**) Percent plaque-GSM variability and (**b**) IM-GSM variability explained by independent predictors. Variables listed in Appendix A that were confirmed with stepwise selection as associated with plaque-GSM or IM-GSM at least 70% of the time in the cross-validation analysis were used as candidates. Among these, the independent predictors were recognized by multiple regression analysis. Partial R^2^ represents the proportion of GSM variability explained by each predictor.

**Figure 4 biomedicines-12-00737-f004:**
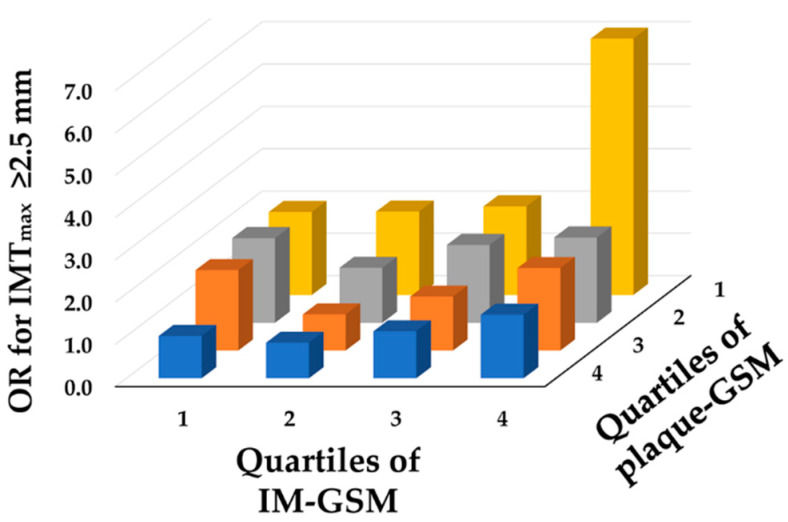
Odds Ratio (OR) for a high atherosclerotic burden (IMT_max_ in the upper quartile) according to quartiles of plaque-GSM and IM-GSM. The more echolucent the plaque (low GSM), the greater the IMT_max_; the more echogenic the IM (high GSM), the greater the IMT_max_. Odds Ratios (OR) of IMT_max_ are calculated after data adjustment for latitude, sex, age, educational level, pulse pressure, and pack-years_code_; and the intersection of the 4th quartile of plaque-GSM and the 1st quartile of IM-GSM was used as a reference. Values of OR and corresponding 95% confidence intervals are given in Appendix A.

**Table 1 biomedicines-12-00737-t001:** Multiple regression analysis showing the independent predictors of carotid plaque-GSM. Partial R^2^ represents the percent variability of the dependent variable explained by each predictor.

	Standardized Beta	*p* Value	Partial R^2^
Latitude	−3.23	<0.0001	8.7%
Diastolic blood pressure	1.36	<0.0001	1.7%
IMT_max_ (quartiles)	−0.96	<0.0001	0.8%
Pack-years_code_	0.85	<0.0001	0.7%
**Whole model**			**12.0%**

**Table 2 biomedicines-12-00737-t002:** Multiple regression analysis showing the independent predictors of common carotid IM-GSM. Partial R^2^ represents the percent variability of the dependent variable explained by each predictor.

	Standardized Beta	*p* Value	Partial R^2^
Latitude	−4.27	<0.0001	11.1%
Height	1.50	<0.0001	2.0%
Systolic blood pressure	1.44	<0.0001	1.5%
PF CC-IMT_mean_ (quartiles)	−1.85	<0.0001	1.2%
Waist/hip ratio	−1.93	<0.0001	1.0%
Fibrates	−0.95	<0.0001	0.6%
Pack-years_code_	0.93	<0.0001	0.6%
MCV	−0.79	<0.0001	0.5%
Alpha-2 inhibitors (sartans)	−0.79	<0.0001	0.5%
Educational level (study years)	−0.79	<0.0001	0.4%
Creatinine (log)	0.96	<0.0001	0.4%
**Whole model**			**19.7%**

## Data Availability

The data presented in this study are available upon request to the corresponding author. The data cannot be made public for ethical reasons.

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
