# Peer review of "Determinants of Carotid Wall Echolucency in a Cohort of European High Cardiovascular Risk Subjects: A Cross-Sectional Analysis of IMPROVE Baseline Data"

_biomedicines, 2024, doi:10.3390/biomedicines12040737_

Round 1
Reviewer 1 Report
Comments and Suggestions for Authors
This is an interesting paper on the association between vascular risk factors and a measure of carotid IM and plaque echogenicity (GSM) conducted in 3,188 subjects in the IMPROVE study. The study showed an overlap in some associated risk factors risk (e.g., latitude, smoking, waist-hip ratio, latitude) but also a distinct risk profile between IM and plaque GSM. Overall, the included risk factors explained about 20% of the variability in IM-GSM and 13% in plaque-GSM, with a significant correlation between them.
This is an interesting study, but several major points need clarifications:
1. How many people had plaque ? How was plaque and IM defined ? Was IM defined in the areas outside plaque? If there is a clear distinction between plaque and IM, a sensitivity analysis among those without plaque may show ‘true’ predictors of IM.
2. Plaque-GSM was obtained from “the darkest plaque” (GSM 0-37) and from echolucent plaque (GSM 104-255). How were these cut offs determined? It would be useful to see the GMS distribution. The analyses treat GSM in a linear model, but both echolucent and echodense plaques are associated with CVD risk (e.g., Yang D, Iyer S, Gardener H, Della-Morte D, Crisby M, Dong C, Cheung K, Mora-McLaughlin C, Wright CB, Elkind MS, Sacco RL, Rundek T. Cigarette Smoking and Carotid Plaque Echodensity in the Northern Manhattan Study. Cerebrovasc Dis. 2015;40:136-43).
3. It is always useful to have a table describing the sample, including demographics, risk factors, SES. The table can be presented by countries. Interestingly, age or other demographics (sex, race) did not seem to be very ‘important’ in these analyses. Can the authors explain.
4. Did the authors include BP, lipid, glucose-lowering medications in the analyses and how did they affect the outcomes?
Thank you.
Reviewer 2 Report
Comments and Suggestions for Authors
My specific comments are listed below:
- line 61: what are the other determinants?
- line 70: what kind of risk factors?
- line 71: what kind of events?
- introduction: what are the limitations of these techniques? are they used anywhere else?
line 96: centers should be named, number of participants from each center should be given
-line 98: please describe in brief
- line 111" please describe in brief
- line 120: whether the automatic measurements were manually verified?
- figure 1: how examples were selected? randomly? most representative?
- line 148: what difference would be unacceptable?
- line 152: how the normality of the distribution was checked?
- line 157: how compliance with the ANOVA conditions was verified?
- line 172: why 70%?
- line 193: what is the meaning of "pack-year score"?
- line 236: no ROC curves presented in the manuscript or supplement information
- figure 4: poor legibility of the figure, it is difficult to relate the individual bars to the OR scale
- line 299: Do the authors have their own suspicions/speculations?
- line 305: Do the authors offer any clinical/practical guidance on when such measurements would be beneficial/necessary?
- line 364: do the authors have any hypotheses about the discrepancy in the observations?
- line 420: whether the observed relationships hold when comparing populations of e.g. exactly 3 VRFs with 4 VRFs?
